# Argon Plasma Treatment Effects on the Micro-Shear Bond Strength of Lithium Disilicate with Dental Resin Cements

**DOI:** 10.3390/ma16155376

**Published:** 2023-07-31

**Authors:** Yixuan Liao, Stephen J. Lombardo, Qingsong Yu

**Affiliations:** 1Department of Mechanical and Aerospace Engineering, University of Missouri, E3411 Lafferre Hall, Columbia, MO 65211, USA; yltpf@mail.missouri.edu (Y.L.); lombardos@missouri.edu (S.J.L.); 2Department of Chemical and Biomedical Engineering, University of Missouri, Columbia, MO 65211, USA

**Keywords:** lithium disilicate, plasma treatment, micro-shear bond strength, surface treatment, dental resin cements

## Abstract

The low bond strength of lithium disilicate (LD) ceramics to dental resin cements remains a critical issue for dental applications because it leads to frequent replacement and causes tooth tissue destruction and consumption. The objective of this study was to examine the effects of atmospheric non-thermal argon plasma (NTP) treatment on LD to improve its micro-shear bond strength (μSBS) with dental resin cements because LD mostly experiences shear stress for its commonly used dental applications as crowns or veneers. Argon plasma treatment was performed on hydrofluoric (HF) acid-etched LD surfaces, and then commercial resin cements were subsequently applied to the treated LD surfaces. The plasma treatment significantly reduced the water contact angle of the LD surface to less than 10° without changing the surface morphology. The μSBS test was performed with cement-bonded LD samples after 24 h and 30 days, as well as after 1000 cycles of thermal cycling. The test results show that, as compared with the untreated controls, 300 s of plasma treatment significantly improved the LD-resin cement bond strength by 59.1%. After 30 days of storage in DI water and 1000 cycles of thermal cycling, the plasma-treated LD samples show 84.2% and 44.8% higher bond strengths as compared to the control samples, respectively. The plasma treatment effect on LD surfaces diminished rapidly as the bond strength decreased to 25.5 MPa after aging in the air for 1 day prior to primer and cement application, suggesting that primers should be applied to the LD surfaces immediately after the plasma treatment. These results demonstrate that, when applied with caution, plasma treatment can activate LD surfaces and significantly improve the SBS of LD with dental resin cements in both short-term and long-term periods.

## 1. Introduction

Lithium disilicate (LD, Li_2_Si_2_O_5_), the ceramic that was introduced into the dental market in the 1990s, can be utilized for tooth- and implant-supported restorations, ranging from single crowns to 3-unit fixed dental prostheses, including anterior veneers to posterior inlays, onlays, and overlays [1]. LD ceramics contain a large fraction (about 70 vol.%) of long crystals, which can improve flexural strength, fracture resistance, and bond strength [2]. There are a number of advantages to this ceramic material, such as excellent esthetics, high strength (400 MPa), versatile applications, and an extensive indication range [3]. In addition, LD is natural-looking, with a similar color to human teeth [4]. To date, LD has been widely used as restorative materials in dentistry, such as inlays, onlays, crowns, veneers, etc. [5,6]. Because of their low porosity and low surface roughness, LD ceramics have a relatively low bond strength to dental resin cements. This low bond strength remains a critical issue for dental applications because it leads to frequent replacement and causes tooth tissue destruction and consumption. Various surface treatments on LD have been studied to improve the bond strength between LD and dental resin cement. Surface treatment methods, including acid etching and/or sandblasting, are often used to create surface micro-irregularities, pits, and roughness to enhance the bond strength of LD to dental resin cements through a micromechanical interlocking mechanism [1]. As a well-established procedure to date, hydrofluoric (HF) acid etching of LD can significantly enhance its bond strength [7]. It is well known that HF is toxic and corrosive, and etching with it can cause excessive loss of the etched material surface [1]. Currently, there is inconclusive data among studies on the preferable method (HF concentration, HF conditioning time, etc.) to pre-treat LD prior to the application of the resin cement [8]. Another commonly used method, sandblasting, can significantly reduce the flexural strength of LD ceramics [9]. In addition, chemical treatment methods like silane pre-treatment of LD prior to the application of a universal adhesive significantly improved its bond strength to dental resin through the formation of strong siloxane linkages [10]. Silane interacts with the silica present in LD and also with the methacrylate molecules present in dental adhesives and resin cements. However, there is limited information with respect to the effectiveness and durability of the bond produced by this technique when applied to LD ceramics [11]. Moreover, it has been reported that there are significant differences between the bond strengths of different commercial composite resin cement systems for LD ceramics [12]. Therefore, it is still necessary to find the appropriate surface treatment methods to improve the shear bond strength of LD to dental resin cement for both short- and long-term stability. In other words, surface treatment of LD surfaces is critical to achieving robust bond strength of dental resin cements for satisfactory dental restorative applications.

Non-thermal plasma (NTP) is a possible surface treatment method to improve the micro-shear bond strength of LD to dental resin cements. Asa novel technology, NTP has been recently applied in dental restoration [13,14,15]. NTP is a partially ionized gas that contains electrons, ions, free radicals, and other reactive particles with energy mainly stored in free electrons; thus, the overall temperature is low so as not to damage the treated material. NTP has been widely used to treat various materials, including ceramics, to improve bonding [16,17,18]. Also, the use of gaseous roughening may make the material surface more uniform. Recently, NTP has exhibited excellent efficacy in oral bacterial deactivation, tooth-whitening treatment, and tooth-composite bonding [19,20,21,22,23,24,25]. Furthermore, it has been confirmed that NTP can improve the bond strength of dental materials, such as enamel and dentin, with different adhesives [26,27,28]. Another study showed that, with 30 s of plasma treatment, a significant increase (by 64%) in bond strength was achieved for composite restoration to peripheral dentin [29]. Plasma treatment also enhanced dentin-adhesive interface bond strength when subjected to different adhesive systems, including total-etch and mild self-etch adhesives [30,31,32]. These results suggest a great potential for NTP to be further used to improve bond strength in many other dental applications. The objective of this study was to investigate the effects of NTP treatment on the surface of LD ceramics and its impact on the micro-shear bond strength between LD and dental resin cement. The study aimed to determine whether NTP treatment could enhance the bond strength of LD to dental resin cement compared to untreated samples. The null hypothesis is that there would be no difference in the micro-shear bond strength between LD samples with no plasma treatment and those subjected to NTP treatment.

## 2. Materials and Methods

### 2.1. Materials and Plasma Device

LD cylinders (R = 0.5 cm, L = 1.0 cm) were acquired from Ivoclar Vivadent (Liechtenstein, Germany). As shown in Table 1, the silane primers used were of three types: ceramic (3M Ceramic Primer, RelyX™ Ceramic Primer, 3M, St. Paul, MN, USA), porcelain primer (Porcelain Primer Silane Coupling Agent, Bisco, Inc., Schaumburg, IL, USA), and bis-silane primer (Bis-Silane^TM^, Parts A and B, Bisco, Inc., Schaumburg, IL, USA). RelyX™ Unicem 2 Automix Self-Adhesive Resin Cement (3M, St. Paul, MN, USA) and Filtek™ Z250 Universal Restorative (3M, St. Paul, MN, USA) composite were used in this study.

Plasma treatment was performed using a lab-made non-thermal plasma brush, with detailed information provided in Ref. [29]. Ultra-high purity argon gas (Industry Grade, Airgas, Radnor Township, PA, USA) was used as the plasma operating gas with a flow rate of 3000 standard cubic centimeters per minute (sccm). An MKS mass flow controller (MKS Instruments Inc., Andover, MA, USA) was used to control the argon gas flow rate. The plasma was formed inside a ceramic chamber and then blown out to form a brush-shaped non-thermal plasma (<40 °C). The plasma brush was operated at a current of 10 mA using a Spellman HV power supply SL60 (Spellman, New York, NY, USA). An optical emission spectroscopy (OES) unit, the Acton 2750 (Princeton Instruments, Trenton, NJ, USA), was utilized to record the light emission from plasma. The OES system (calibrated using IntelliCal™) has a grating of 150 grooves per millimeter with a blazing wavelength of 500 nm. Spectra were acquired from 200 nm to 900 nm in step and glue modes.

### 2.2. Sample Preparation

LD cylinders were mounted in an acrylic resin prepared from a 2:1 ratio of QuickCure Acrylic powder (Allied, CA, USA) and QuickCure Acrylic liquid (Allied, CA, USA). The LD embedding mold was stored in an ice-water bath for 1 h after the resin solution was introduced into the mold. The mounted samples were extracted from the mold and stored in de-ionized (DI) water at 4 °C until use. The LD surface was first sanded flat by a Trimmer machine (JT19, Zeny, Fontana, CA, USA) and then polished with 600-grit sandpaper (Norton Abrasives, Worcester, MA, USA).

Figure 1 shows the preparation procedure for the LD specimens. The LD surface was first etched using HF acid gel (Ivoclar, Mississauga, ON, Canada) for 20 s, then thoroughly rinsed with DI water for 30 s, and air-dried. For the control group without plasma treatment, silane primers were applied to the LD surface and gently dried with compressed air. To ensure the amount of resin used for the sample preparation, a mold made by a 10-layer tape with a punched hole (diameter: 6.0 mm) was then attached to the LD surface to expose the LD with a surface area of $A=\pi R^{2}=$ 28.3 mm^2^ for the silane primer application. Then, a thin layer (~0.45 mm thick) of resin cement was applied to the primer-coated LD surface and light-cured (Superdental LED light, North Andover, MA, USA) for 10 s, with excessive resin cement cautiously removed using a razor blade. The sample was fixed on the Ultradent mold (Bonding Clamp and Bonding Mold Inserts, Ultradent Products Inc., South Jordan, UT, USA) with the composite applied through the mold and cured using a dental light (Superdental LED light, North Andover, MA, USA) for 10 s.

Figure 2 shows the preparation procedure and sample designation. Plasma treatment was performed on the air-dried HF acid-etched LD surface for different times (t = 30 s, 60 s, 90 s, 120 s, and 300 s). The rest of the bond procedures for the sample preparation were the same as for the control samples without plasma treatment. 15 test specimens were prepared and tested for each group. All the test specimens were stored in DI water at 37 °C for 24 h according to ISO/TS 11405 before proceeding to the micro-shear bond strength (μSBS) test [33]. The long-term storage groups were further divided into 30-day storage groups and thermal cycling groups. The 30-day storage groups were held for 30 days of storage in DI water at 37 °C. Thermal cycling was performed by alternatingly storing the specimens between two water baths with temperatures of 5 °C and 55 °C for 1000 cycles.

The effect of aging on plasma-treated LD surfaces was further examined in terms of water surface contact angle and micro-shear bond strength. Water contact angle measurements were performed on plasma-treated LD surfaces at different times, from 5 s to 24 h (1 day) after the plasma treatment. Similarly, the test specimens for the micro-shear bond strength test were prepared with the plasma-treated LD at different times, from 5 s to 24 h (1 day) after the plasma treatment.

### 2.3. Micro-Shear Bond Strength (μSBS)

The μSBS tests were performed using an Instron universal testing machine (Instron 3367 Dual Column Testing Systems, Instron, MA, USA) with Bluehill software (Bluehill 2, Instron, MA, USA). A test specimen was placed in the test base clamp, and a shear force was applied to each specimen at a crosshead speed of 1 mm/min until failure occurred. The maximum force was recorded by the software. The fractured surfaces were examined to determine the failure mode using an optical microscope (AmScope NMM-800TRF, AmScope, Irvine, CA, USA). The bond failure modes were classified into three types: adhesive (failure occurred at the interface between LD and resin cement), cohesive (failure occurred within the resin composite), and mixed (adhesive + cohesive failure both exist on the substrates).

### 2.4. Water Contact Angle

Water contact angles were measured using a computer-aided VCA 2500 XE Video Contact Angle System (AST Products, Inc., Billerica, MA, USA) on a 0.5 μL droplet of distilled water placed onto the LD surface. Software Image J was used to analyze the water contact angles.

### 2.5. Surface Morphology 

The LD surfaces were examined and analyzed using a field-emission scanning electron microscope (SEM) (Philips XL30, FEI, Hillsboro, OR, USA). Untreated LD, HF acid-treated LD, and HF acid-etched LD with 300 s of plasma treatment time were prepared and then sputter coated with platinum before the SEM measurement. An optical profilometer (Veeco NT 9109, Veeco Model, Plainview, NY, USA) was used to measure the surface roughness of LD, HF acid-treated LD, and HF acid-etched LD with 300 s plasma treatment.

### 2.6. Statistical Analysis

Statistical analyses were performed using OriginLab (OriginLab Software). The μSBS results were presented as the mean standard deviation (SD). ANOVA and Tukey’s test were used to compare different bond groups. The mean differences between groups were considered significant at *p* < 0.05 for all analyses. (* represents *p* < 0.05, ** represents *p* < 0.01, *** represents *p* < 0.001, and **** represents *p* < 0.0001). The Chi-square test was used to analyze the significant difference in failure mode among experimental groups. 

## 3. Results

Figure 3 shows the μSBS results of plasma-treated LD-resin cement test specimens prepared using 3M Ceramic Primer along with the untreated control group, which is noted as having a plasma treatment time of 0 s. The LD-resin cement bond strength for the untreated control group was 32.6 ± 5.7 MPa. With plasma treatment, improvements in bond strengths were achieved. With plasma treatment time increasing, the LD-resin cement bond strength increased. A significant increase in bond strength was achieved for the plasma-treated groups with plasma treatment time variation from 30 s to 300 s as compared with the control group’s 0 s treatment time. The highest bond strength achieved was 51.9 ± 4.0 MPa, which increased by 59.1% as compared to that of the control group without plasma treatment. These results indicate that the plasma treatment can effectively enhance the bond strength between LD and dental resin cement using ceramic primer. The longer the plasma treatment was applied to LD surfaces, the greater the improvement in bond strength was achieved. 

Figure 4 shows the μSBS results of LD-resin cement test specimens prepared using Bisco Porcelain primer along with the untreated control group (t = 0 s), which is noted as the plasma treatment time of 0 s. The bond strength of the untreated group was 27.6 ± 4.9 MPa. The bond strength of the plasma-treated sample groups increased. With 120 s plasma treatment, the LD-resin cement bond strength obtained the highest bond strength for the porcelain primer group with 32.0 ± 7.3 MPa, which increased by 23.9% as compared to the untreated HF acid etched experiment group. Statistical analysis, however, indicated no significant difference between the plasma-treated groups and the untreated group without plasma treatment. While plasma treatment provided an improvement in the micro-shear bond strength of LD to dental resin cement by using Bisco porcelain primer, the difference was not statistically significant compared to the ceramic primer. To further test the plasma treatment effect, an additional primer from Bisco was applied and examined.

Figure 5 displays the μSBS results of the LD-resin cement test specimens prepared using Bisco Bis-Silane primer along with the untreated control group (t = 0 s). The bond strength of the untreated group was 26.2 ± 5.2 MPa. With plasma treatment, significantly higher bond strengths were achieved for all the plasma-treated groups. With 300 s of plasma treatment, the LD-resin cement bond strength of the Bis-silane primer group increased to 35.7 ± 6.2 MPa, which was 36.2% higher than the untreated control group. From the one-way ANOVA, the bond strength between LD and resin cement using the Bisco Bis-silane primer showed a significant difference (*p* = 0.003) between the control group and plasma treated groups. The mean comparison further confirmed that the bond strength enhancement with 300 s plasma treatment was statistically significant as compared with the control group without plasma treatment (*p* = 0.00058). These results indicate that plasma treatment of LD significantly improves its micro-shear bond strength to dental resin cement when Bisco Bis-silane primer was used for the sample preparation.

Figure 6 shows the μSBS test results of the LD-resin cement test specimens after 30 days of storage in DI water. It gives a comparison between the plasma-treated groups and their corresponding controls without plasma treatment. It was noted that, with 300 s of plasma treatment, the bond strength of LD-resin cement test specimens prepared with all three silane primers showed higher values as compared to their controls without plasma treatment. When Bisco porcelain primer was used, the 300-s plasma-treated group had a bond strength of 28.7 ± 4.6 MPa, which is 84.2% higher than its corresponding control group without the plasma treatment (15.6 ± 4.9 Mpa). This improvement was statistically significant (*p* < 0.0001) as compared with the control group. 

The μSBS test was also performed with LD-resin cement test specimens after 1000 cycles of thermocycling between 5 °C and 55 °C. The comparison is made between the 300-s plasma-treated groups and their corresponding control groups without plasma treatment. As shown in Figure 7, the bond strength of the plasma-treated groups all significantly increased (*p* < 0.0001) as compared with their respective control groups without plasma treatment. The 300-s plasma treatment group prepared using 3M Ceramic primer showed a bonding strength of 31.5 ± 5.1 MPa as compared with the 21.8 ± 4.0 Mpa obtained from its control group. When Bisco porcelain primer was used, the 300-s plasma treatment group had a bonding strength of 28.0 ± 4.5 Mpa as compared with 18.1 ± 3.8 Mpa in its control group. When Bisco Bis-silane primer was used, the 300-s plasma treatment group had a bonding strength of 33.5 ± 4.1 Mpa as compared with the 24.0 ± 2.9 Mpa of its control group. After 1000 thermal cycles, all the sample groups with 300 s plasma treatment showed significantly higher (*p* < 0.0001) bonding strengths as compared to their corresponding control groups, indicating that the LD-resin cement bonding was much more durable with plasma treatment as compared to without. The plasma treatment did improve the micro-shear bond strength and enhance the durability of the LD-resin cement bonding performance.

Figure 8 shows the SEM images of the LD surfaces after different surface modifications. Without any treatment, the LD surface was relatively smooth, with some scattered micro-dents and some scratches resulting from the sandpaper polishing. After HF acid etching, the LD surface became significantly rougher, with uniformly distributed and much deeper dents. With further plasma treatment on the HF acid-etched LD surface, no obvious changes in surface morphology were observed, indicating that the plasma treatment did not have a noticeable impact on the LD surface morphology. To further evaluate the surface morphology, an optical profilometer microscope was also utilized. The surface morphology of LD, HF acid-etched LD, and HF acid-etched LD with 300 s plasma treatment were studied. From Figure 9 and Table 2, the surface roughness of LD increased after HF acid etching. In contrast, plasma treatment on HF-etched LD did not significantly alter the surface roughness, which was consistent with the observations from SEM images. These findings demonstrate that HF acid etching significantly increases the surface roughness of LD. However, subsequent plasma treatment does not affect the LD surface roughness.

Figure 10 shows the water contact angle change on the no-HF acid-etched LD surfaces without plasma treatment and with plasma treatment at different times. It can be seen from Figure 10 that the HF acid-etched LD surface is very hydrophilic, with a water contact angle of ~17°. After plasma treatment, the water contact angles on the LD surfaces continuously decreased with plasma treatment time, indicating that the LD surfaces became more hydrophilic, and the surface energy increased with increasing plasma treatment time.

As shown in Figure 11A, the plasma-treated LD surfaces showed rapid hydrophobic recovery after plasma treatment. It can be seen that, with 1 day of aging, the plasma-treated LD surfaces had a water contact angle similar to the LD surfaces without plasma treatment. To further evaluate the aging effect on the bond strength, LD-resin cement test specimens were prepared using 3M Ceramic primer on plasma-treated LD surfaces with aging times of 5 s, 1 min, 2 min, 5 min, 10 min, 1 h, 2 h, 5 h, and 1 day. The LD-resin cement test specimens prepared within 5 s of aging time after plasma treatment showed a significantly improved bond strength (see Figure 11B) of 51.9 ± 4.0 MPa as compared to 32.6 ± 5.7 MPa measured for the control group without plasma treatment. However, the LD-resin cement test specimens prepared with aging time ≥ 1 min after plasma treatment) showed bond strength similar to the 32.6 ± 5.7 MPa measured in the control group without plasma treatment. This result suggests that the primer should be applied immediately after plasma treatment to prevent rapid hydrophobic recovery.

After μSBS tests, the fracture surfaces of all the LD-resin cement test specimens were further examined under an optical microscope to identify the three bond failure modes: adhesive (failure occurred at the interface between LD and resin cement), cohesive (failure occurred within the resin composite), and mixed (adhesive + cohesive failure both exist on substrates). Figure 12 shows the distribution of the failure modes of the tested LD-resin cement specimens after 24 h of storage in 37 °C DI water. It can be noted that, after 24 h of storage in DI water, the adhesive failure dominated for the test specimens prepared using all three silane primers (3M Ceramic, Bisco Porcelain, and Bisco Bis-silane). In contrast, Figure 13 shows that after 30 days of storage in 37 °C DI water or 1000 cycles of thermal cycling, the number of cohesive failure modes increased for the test specimens prepared using Bisco Porcelain and Bis-silane primers, indicating improved bonding durability at the LD-primer interface.

The non-thermal atmospheric argon plasma used in this study was characterized using OES. As shown in Figure 14A,B, the majority of the optical emissions can be assigned to Ar emission in the wavelength range from 680 to 900 nm, along with much weaker N_2_ emissions from 300 to 450 nm. Oxygen atomic emission lines were also detected around 777 nm and 844 nm, as shown in Figure 14C,D. The presence of nitrogen and oxygen emission lines indicates that the argon plasma interacted with the ambient air and generated reactive oxygen species (ROS) and reactive nitrogen species (RNS).

## 4. Discussion

The null hypothesis of this study was that there would be no significant performance difference between sample groups treated with and without plasma. Based on the results herein, the null hypothesis was rejected. As shown in Figure 3, Figure 4, Figure 5, Figure 6 and Figure 7, with plasma treatment, the micro-shear bond strength of the LD-resin cement samples showed a significant improvement compared to the control group without plasma treatment. For 3M Ceramic primer, the bond strength of LD-resin cement increased by 59.1% with 300 s plasma treatment compared to the control group of 0 s plasma treatment. With plasma treatment, the LD-resin cement bonding strength increased to 23.9% and 36.2% for Bisco porcelain primer and Bis-silane primer, respectively.

To evaluate the longevity of the bond of LD-resin cements, μSBS tests were performed on test specimens after 30 days of storage in DI water and after 1000 cycles of thermocycling. As shown in Figure 6, after 30 days of water storage, the bond strength of plasma-treated specimens increased as compared with the control specimens without plasma treatment. When Bisco porcelain primer was used, the plasma-treated group showed a significantly higher bond strength than the control group without plasma treatment. When 3M ceramic primer and Bisco bis-silane primer were used, however, no significant differences were observed between the plasma treatment groups and the corresponding control groups. Such results could be due to the pH value difference between the three primers. The pH values of 3M ceramic primer, Bisco porcelain primer, and Bisco bis-silane primer are 4.6, 5.9, and 4.0, respectively. It was reported that silane hydrolysis is faster in acidic conditions [34]. The Bisco Porcelain primer has the highest pH value, indicating that silane hydrolysis is slower than the other two primers and takes more time to complete hydrolysis to bond onto LD surfaces. As shown in Figure 4, after 24 h of storage in DI water, the bond strength of LD-resin cements using Bisco Porcelain primer was lower than using the other two primers shown in Figure 3 and Figure 5. After 30-day storage in DI water and after 1000 cycles of thermocycling, the porcelain primer completed the hydrolysis process and resulted in a similar bond strength of LD-resin cement to the specimen group using 3M Ceramic and Bisco Bis-silane primers.

The thermocycling process simulates the long-term consumption of restorative materials in the oral environment [35,36]. As shown in Figure 7, after 1000 cycles of thermocycling, the plasma-treated specimen groups using all 3 silane primers all exhibited significantly increased LD-resin cement bond strength. This result demonstrated that plasma treatment improved the bond strength of LD-resin cement and thus the bond’s longevity. However, some studies used 5000 and 10,000 cycles of thermocycling to represent the longer clinic service [37,38]. Extended thermocycling can simulate the stress and temperature changes over a longer period in an oral environment, providing a more realistic assessment of the bond strength and performance of the dental restorative material over time. Including more thermocycling cycles in future studies can further validate and strengthen the findings regarding the long-term effectiveness of the plasma treatment on LD-dental resin cement bonding. 

One of the important factors that may influence the bond strength of LD-resin cement is the wettability of the LD surface with primers. The ideal wetting situation is to cover the LD substrate completely to maximize the contact of an adhesive to the substrate [39,40,41]. A hydrophilic substrate surface with high surface energy is most often desired for a good bond [42]. After plasma treatment, the LD surface became more hydrophilic as the water contact angle decreased from ~17° to almost ~6°. The lower contact angle indicates higher surface energy, which leads to improved micro-shear bond strength in LD-resin cement samples.

Surface morphology is another important factor that affects the bond strength between the LD and resin cement. In this study, the LD surface showed no surface morphology change from before to after plasma treatment, indicating that the plasma treatment has no effect on the surface roughness of the LD but just increases the surface energy of the LD. Similar results were found in previous studies [41,43], which showed that plasma treatment has no influence on the substrate surface morphology. Prior to the plasma treatment, hydrofluoric (HF) acid etching was applied as a pre-treatment of the LD surface. It is well known that HF acid has the ability to selectively dissolve the crystalline components of ceramic materials, thus producing a porous and irregular surface, increasing surface area, and improving resin penetration into the etched ceramic surface with micro-retentions [12]. HF acid etching can change the LD surface topography with increased roughness for adhesive bonding by removing or stabilizing surface defects [44]. However, some studies have shown that HF acid etching has the potential to form cracks, which may interfere with the adhesion of resin cements and therefore reduce the bond stability of LD-resin cement [45,46,47]. To minimize these damaging effects of HF acid etching on LD surfaces, subsequent surface treatment using silane coupling agents could ensure a more stable and improved bond strength of LD to resin cements [48,49,50]. 

Silane coupling agents play an important role in effectively promoting the adhesion of silica-based materials. Application of silane coupling agents to ceramic surfaces provides more chemically covalent and hydrogen bonding between the resin system and the ceramic surfaces [51] and thus sufficient bond sites for the resin to bond with [52,53,54]. Generally, silane coupling agents can react with water and form free silanol groups that can then bond with the hydroxyl groups on the LD surfaces. Resin cements contain methacrylate groups that can react with silane, therefore, a strong linkage between LD and resin cement was formed with the assistance of silane coupling agents [55]. Plasma treatment activated LD surfaces, promoted the covalent and hydrogen bonds of silane, and thus improved the LD bond strength to resin cements, as observed in this study.

The stability of the plasma-treated LD surfaces was studied in terms of water contact angle change and bond strength change with aging time. After plasma treatment, a fast hydrophobic recovery was observed within 10-min of aging. As shown in Figure 11B, a significantly enhanced micro-shear bond strength was achieved with LD-resin cement specimens when 3M Ceramic primer was applied to the LD surfaces within 5 s after plasma treatment. In contrast, the bond strength decreased rapidly with the aging time before the primer was applied after plasma treatment. When the primer was applied beyond 10 min after plasma treatment, the bond strength of the LD-resin cement was similar to that of the control group without plasma treatment. When plasma treatment is applied to the LD surface, it introduces a large number of highly energetic argon ions, electronically excited metastable atoms, and some reactive oxygen species (ROS) that come into contact with the LD ceramic surface. The impact of the highly energetic argon plasma species and the presence of ROS leads to an increase in surface energy and hydroxy groups on the LD surface, making it more hydrophilic and enhancing its surface reactivity. As a result, the LD surface becomes more receptive to bonding with the functional monomer present in the primer. However, if no primer is applied immediately after plasma treatment, the amount of polar hydroxy groups on the LD surface may start to decrease, as an observed hydrophobic recovery with aging time is shown in Figure 11A. This can lead to changes in the LD surface properties and reduce the effectiveness of the plasma treatment in activating the surface for bonding with dental resin cements. These results indicate that plasma treatment effects on LD surfaces diminish rapidly after plasma treatment. It is suggested, therefore, that silane primers need to be applied to LD surfaces immediately after the plasma treatment to ensure enhanced bond strength.

As a partially ionized gas created through electric power, non-thermal argon plasmas contain various energetic and reactive species, including energetic free electrons, ions, and electronically excited metastable atoms or molecules. As shown in Figure 14, the OES spectrum of the argon plasmas showed strong photoemissions from electronically excited Ar atoms with high emission intensity. Weak emission lines and bands were also observed for electronically excited O atoms and electronically excited N_2_ molecules, which are commonly noted as ROS and reactive nitrogen species (RNS), respectively, formed through plasma reactions with oxygen and nitrogen in ambient air. The energetic free electrons, Ar ions, electronically excited Ar atoms, ROS, and RNS in the plasma could inject more energy onto LD surfaces through bombardment. In addition, ROS can form active peroxide radicals to initiate chemical surface changes on inert ceramic surfaces, including LD surfaces. The surface activation effect of the plasma treatment was reflected in the significant reduction of the water contact angle, i.e., the surface energy increase on the LD surfaces.

In the dental restorative material bond test, shear, μSBS, tensile, and micro-tensile bond strength (μTBS) were the four popular tests [56]. μTBS tests were used because LD mostly experiences shear stress for its commonly used dental applications, such as crowns or veneers. In μSBS test, a shear force is applied parallel to the bonded interface, mimicking the forces experienced during chewing and other oral functions. This test provides a more uniform stress distribution along the bonded interface, reducing stress concentration and potentially leading to more accurate bond results. On the other hand, μTBS test applies the force perpendicular to the surface of the substrate. This test method may result in stress concentration at the gripping points of the specimen, which could potentially influence the overall results [57]. However, it is worth considering conducting μTBS tests in future studies to analyze any differences or potential variations in results compared to the μSBS tests. Ultimately, using both test methods can provide a more comprehensive understanding of the bond strength characteristics of dental restorative material.

In this study, non-thermal plasma treatment had a positive effect on improving the micro-shear bond strength of LD to dental resin cement, both in the short-term and long-term. The plasma treatment significantly increased the hydrophilicity of the LD surface without causing any surface morphology variation, indicating its potential as an effective bond strength improvement technique for dental restorative materials.

## 5. Conclusions

As a restorative material used in dentistry, LD has low bond strength and limited short-term and long-term bond stability with dental resin cements. In this study, non-thermal atmospheric pressure argon plasma treatment was used to activate LD surfaces to improve their shear bond strength with dental resin cements. With increasing plasma treatment time, a significant 59.1% increase in the short-term micro-shear bond strength of LD-resin cements was obtained after 24 h of sample storage. The plasma treatment also provided a significant increase of 44.8% and 84.2% in long-term shear bond strength after 1000 cycles of thermocycling and after 30 days of storage in 37 °C DI water, respectively, as compared to the control groups without plasma treatment. Without affecting surface morphology, plasma treatment of LD surfaces reduced their water contact angle, indicating an increase in surface hydrophilicity, wettability, and surface energy. It was also found that the plasma treatment effect diminished rapidly if the silane primer and resin cement were not applied immediately after plasma treatment. In summary, NTP surface treatment is a promising technology for improving the micro-shear bond strength of LD to dental resin cements for short- and long-term stability, but it needs to be applied with caution to retain and benefit from the plasma surface activation effects. Further studies are expected to optimize the clinical applicability of NTP in dental restorative applications.

## Figures and Tables

**Figure 1 materials-16-05376-f001:**
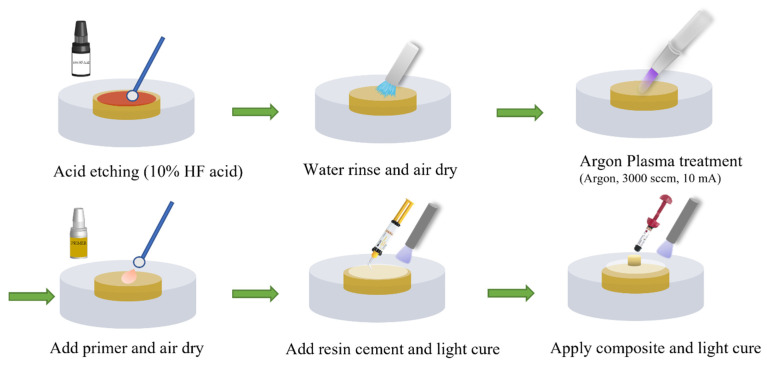
Schematic of the lithium disilicate sample preparation procedure.

**Figure 2 materials-16-05376-f002:**
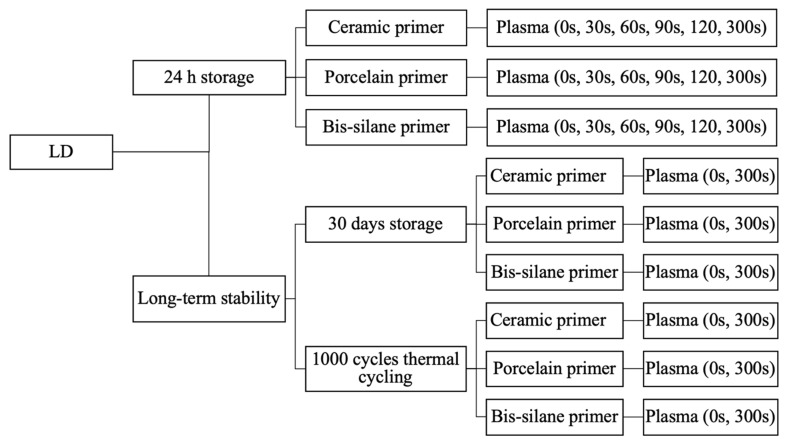
Lithium disilicate sample groups for the micro-shear bond strength test.

**Figure 3 materials-16-05376-f003:**
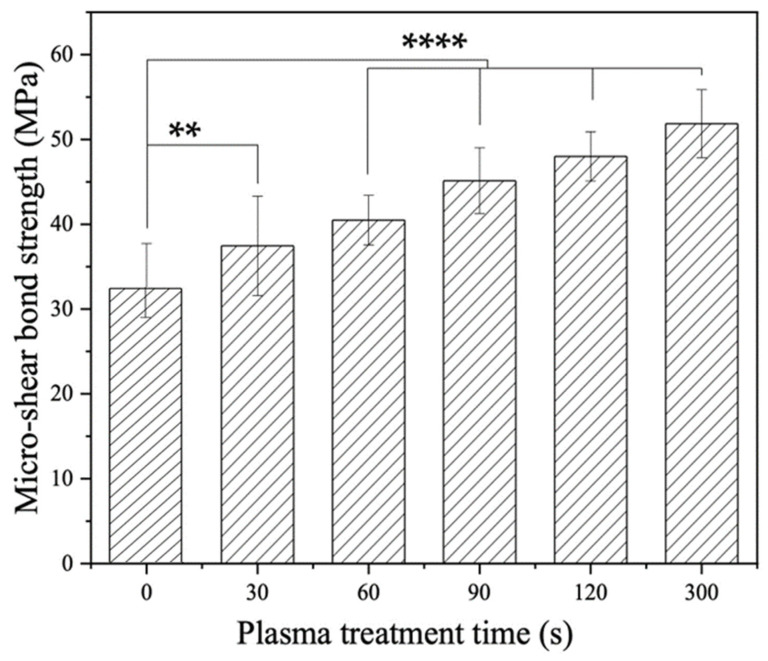
Micro-shear bond strength of LD-resin cement specimens prepared with 3M Ceramic primer versus plasma treatment duration, with 0 s being the untreated control. The test was performed on test specimens stored in 37 °C DI water for 24 h. (ANOVA: *p* = 0, Tukey mean comparison test: ** represents *p* < 0.01 and **** represents *p* < 0.0001).

**Figure 4 materials-16-05376-f004:**
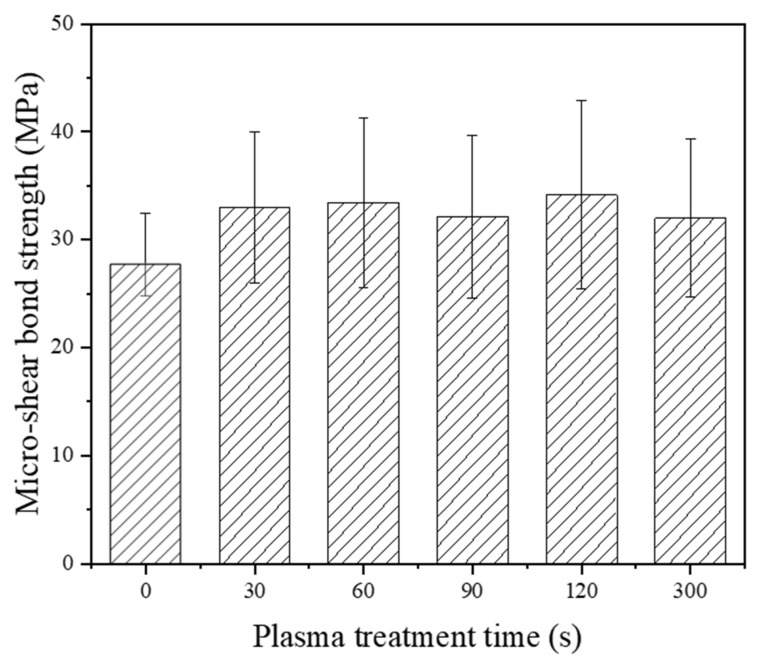
Micro-shear bond strength of LD-resin cement specimens prepared with Bisco Porcelain primer versus plasma treatment duration with 0 s being the untreated control. The test was performed on specimens stored in 37 °C DI water for 24 h. (ANOVA: *p* = 0.189).

**Figure 5 materials-16-05376-f005:**
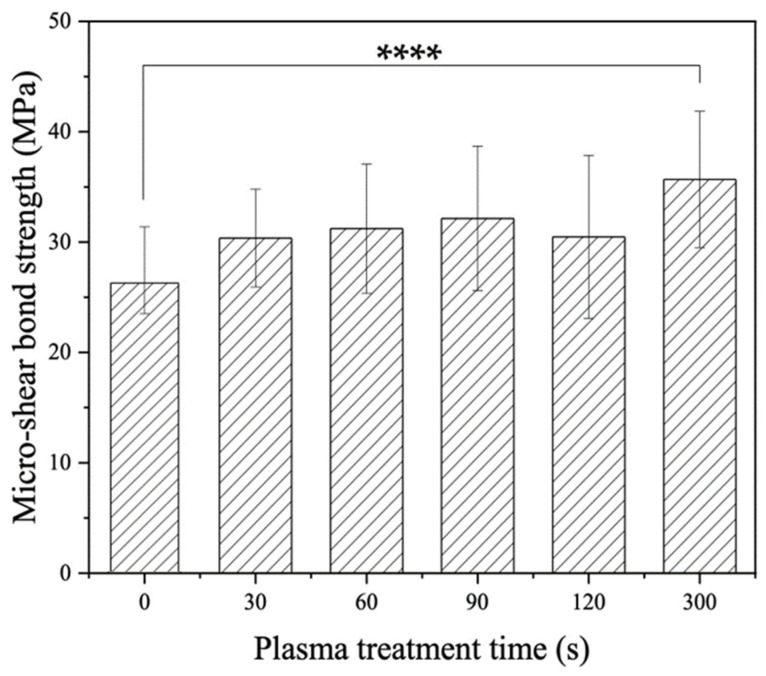
Micro-shear bond strength of LD-resin cement specimens prepared with Bisco bis-silane primer versus plasma treatment duration, with 0 s being the untreated control. The test was performed on specimens stored in 37 °C DI water for 24 h. (ANOVA: *p* = 0.003, Tukey mean comparison test: **** represents *p* < 0.0001).

**Figure 6 materials-16-05376-f006:**
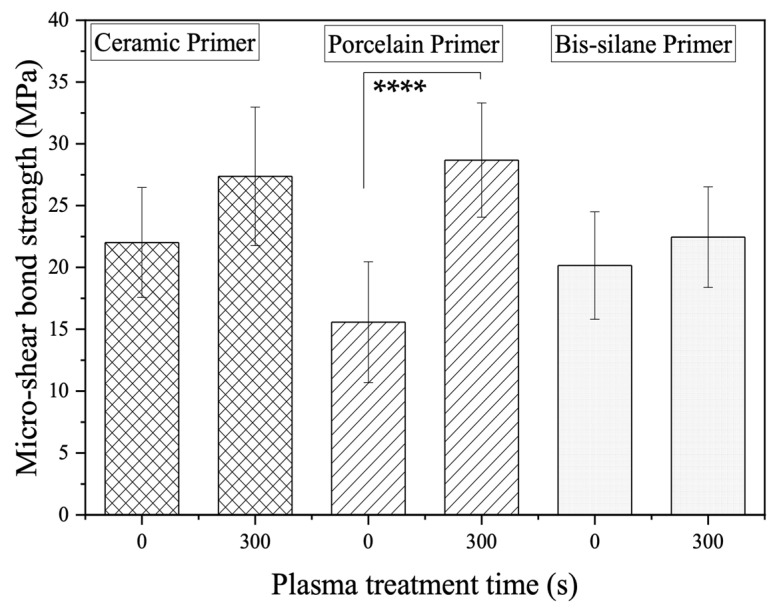
Comparison of the micro-shear bond strength of the untreated control specimens without (0 s) and with (300 s) of plasma treatment for LD-resin cement specimens prepared with three primers. The test was performed on specimens after storage in 37 °C DI water for 30 days. (ANOVA: *p* = 0.029 for ceramic primer, *p* = 8.07 × 10^−6^ for porcelain primer, *p* = 0.238 for bis-silane primer, and **** represents *p* < 0.0001).

**Figure 7 materials-16-05376-f007:**
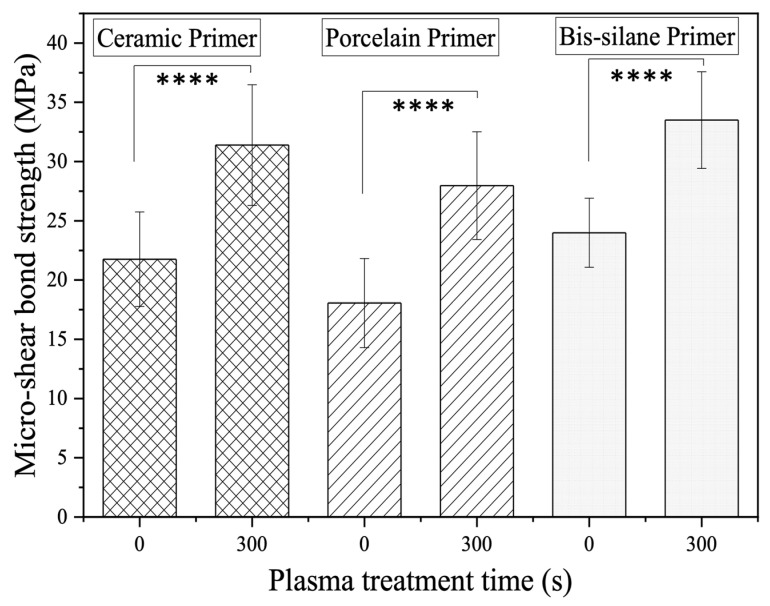
Comparison of micro-shear bonding strength of the untreated control specimens without (0 s) and with 300 s of plasma treatment for LD-resin cement specimens prepared with three primers. The test was performed on specimens after 1000 thermocycles. (ANOVA: *p* = 3.388 × 10^−6^ for ceramic primer, *p* = 4.686 × 10^−7^ for porcelain primer, *p* = 5.262 × 10^−8^ for bis-silane primer, and **** represents *p* < 0.0001).

**Figure 8 materials-16-05376-f008:**
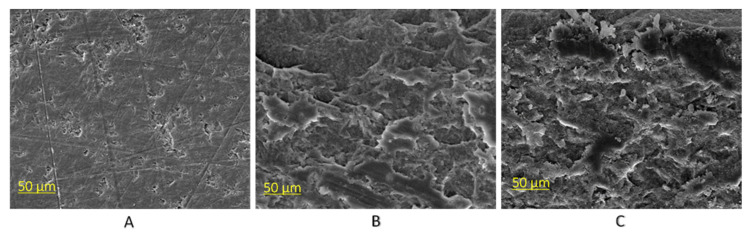
SEM images of (**A**) the LD surface after polishing with sandpaper; (**B**) the LD surface after HF acid etching; and (**C**) the LD surface after HF etching followed by 300 s plasma treatment.

**Figure 9 materials-16-05376-f009:**
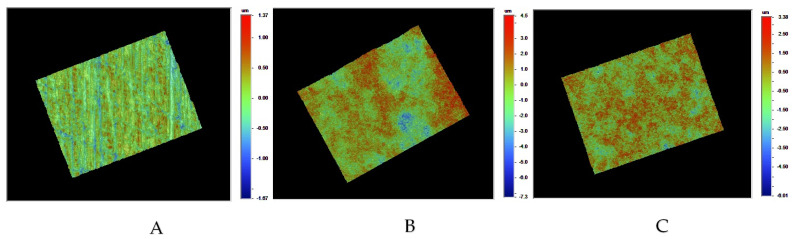
Surface morphology images of (**A**) the LD surface after polishing with sandpaper; (**B**) the LD surface after HF acid etching; and (**C**) the LD surface after HF etching followed by 300 s plasma treatment.

**Figure 10 materials-16-05376-f010:**
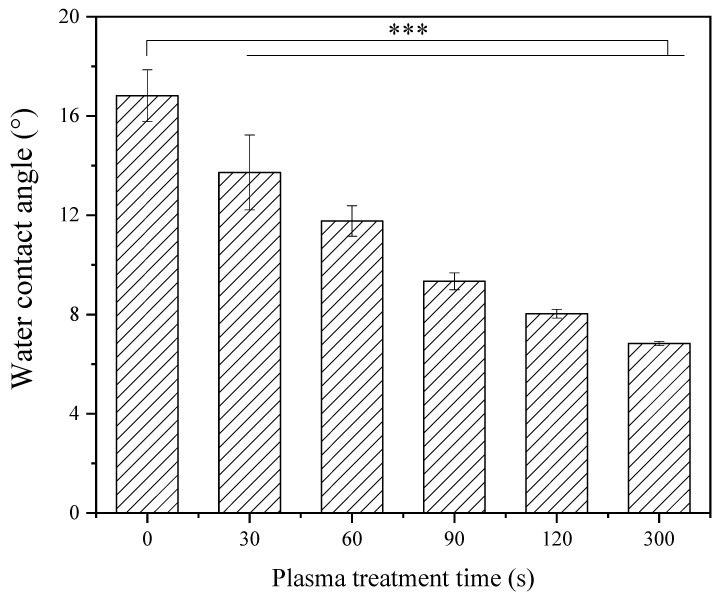
Water contact angle on no HF acid etched LD surfaces without plasma treatment (0 s) and with plasma treatment for different time from 30 s to 300 s. (*** represents *p* < 0.001).

**Figure 11 materials-16-05376-f011:**
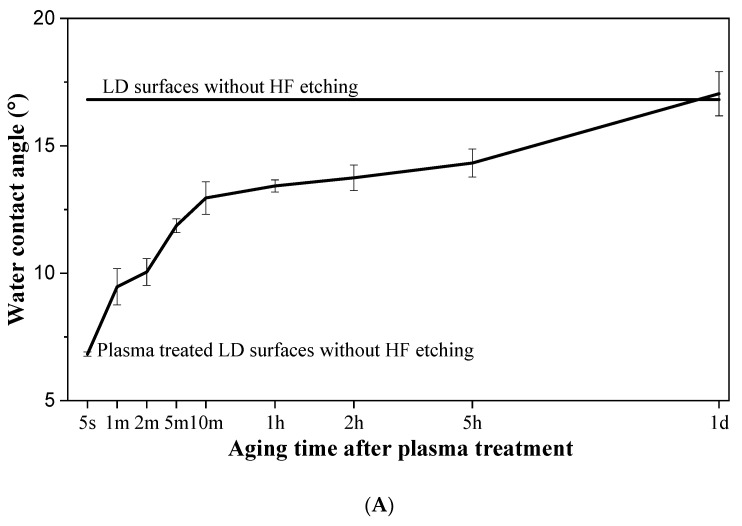
(**A**) Water contact angle change with the aging time of plasma treated LD surfaces, and (**B**) micro-shear bond strength of the test specimens prepared at different aging times of plasma treated LD surfaces using 3M Ceramic primer (**** represents *p* < 0.0001). Plasma treatment time was 300 s.

**Figure 12 materials-16-05376-f012:**
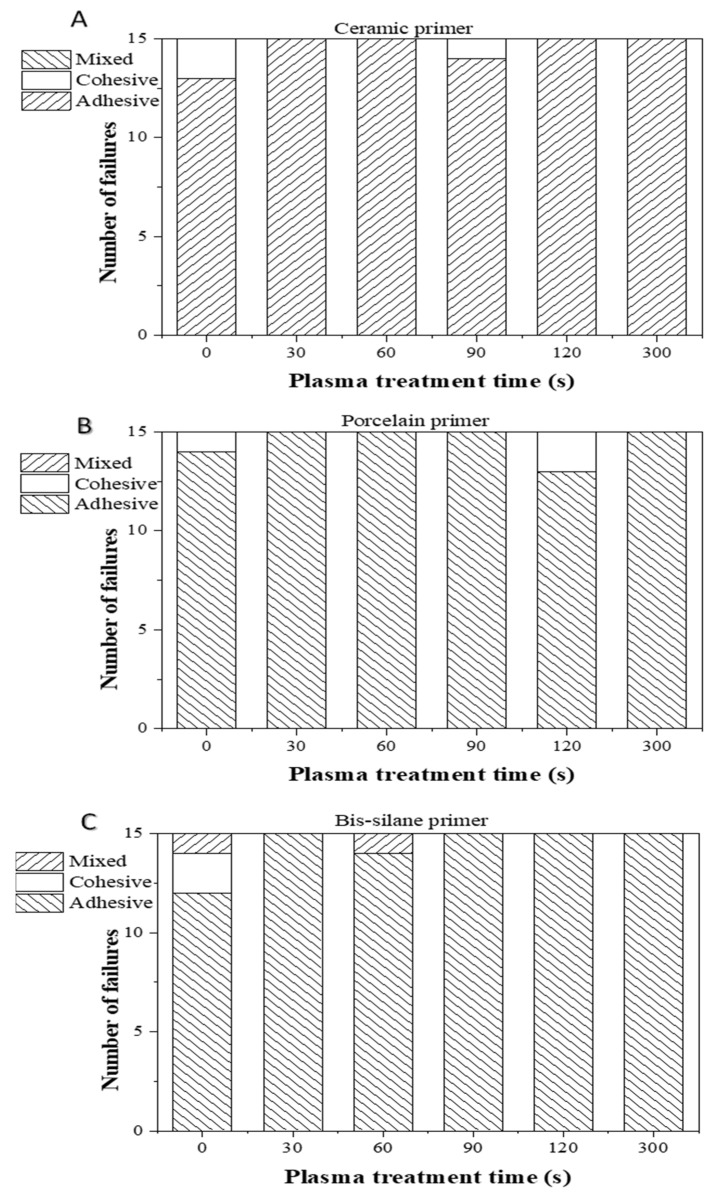
Micro shear bond failure mode of LD-resin cement test specimens prepared using (**A**) ceramic primer, (**B**) porcelain primer, and (**C**) bis-silane primer. The test was performed after the specimens were stored in 37 °C DI water for 24 h.

**Figure 13 materials-16-05376-f013:**
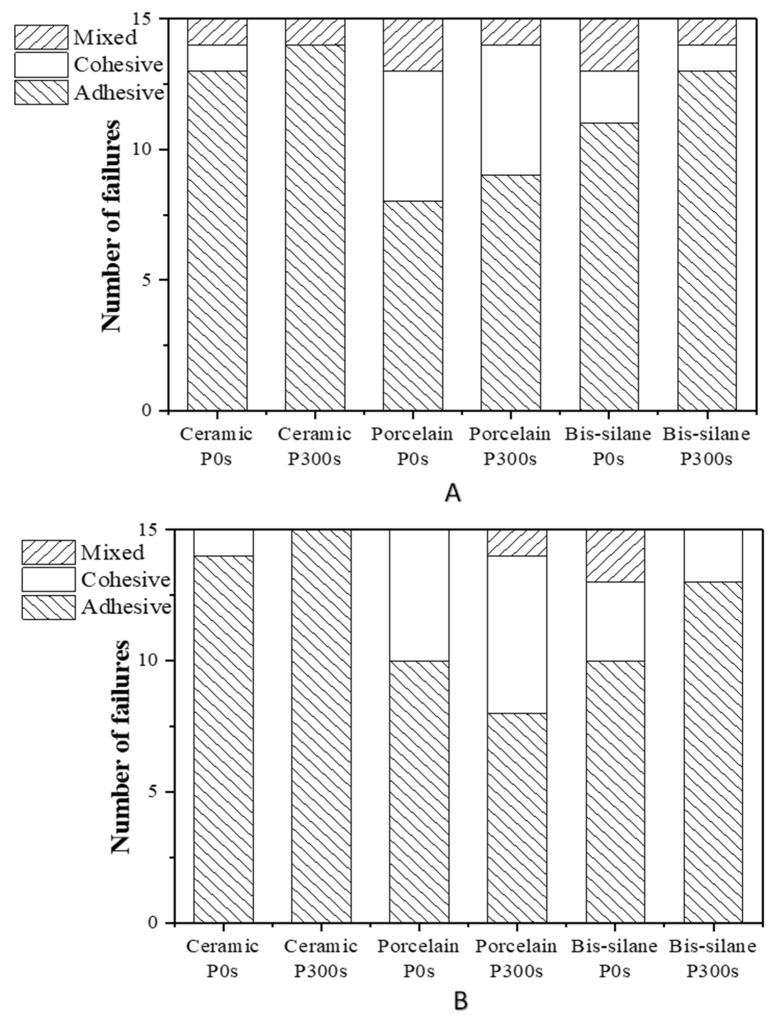
Micro shear bond failure modes after LD-resin cement test specimens prepared using three different primers after (**A**) 30 days of storage in 37 °C DI water and (**B**) 1000 thermocycles. P0s indicates no plasma treatment, and P300s indicates plasma treatment for 300 s.

**Figure 14 materials-16-05376-f014:**
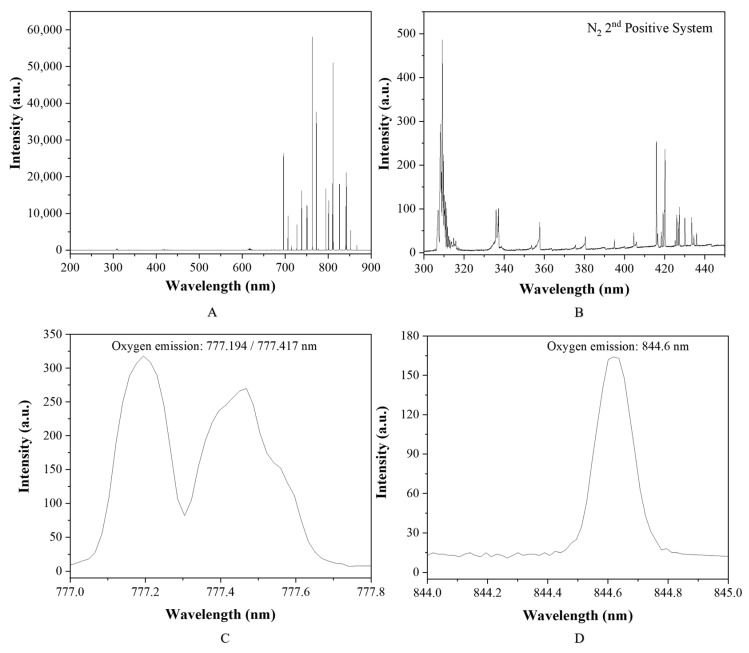
OES spectrum of the argon plasma: (**A**) the 300–900 nm wavelength region showing strong argon emission; (**B**) the 300–450 nm wavelength region showing nitrogen in the second positive system of the argon plasma; (**C**) weak oxygen emission between 777–778 nm; (**D**) weak oxygen emission at 844–845 nm.

**Table 1 materials-16-05376-t001:** The silane primer coupling agents used in the study.

Product Name	Ingredient (% by wt)
3M ESPE RelyX Ceramic Primer	Ethyl alcohol (70–80%), Water (20–30%), Methacryloxypropyltrimethoxysilane (<2%)
Bisco Porcelain Primer	Ethanol (30–50%), Acetone (30–50%), Silane (1–5%)
Bisco Bis-silane Primer	Part A: Ethanol (>85%), 3-(Trimethoxysilyl) propyl-2-Methyl-2-Propenoic Acid (5–10%) Part B: Ethanol (30–50%), Phosphoric Acid conc = 85% (1–5%)

**Table 2 materials-16-05376-t002:** Surface roughness data of LD measured using profilometer.

LD Surface	*Ra* (nm)	*Rq* (nm)
After polishing with sandpaper	189 ± 41	247 ± 46
After HF acid etching	883 ± 139	1094 ± 176
After HF etching followed by 300 s plasma treatment	940 ± 222	1238 ± 224

## Data Availability

No applicable.

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
