# Peer review of "Argon Plasma Treatment Effects on the Micro-Shear Bond Strength of Lithium Disilicate with Dental Resin Cements"

_materials, 2023, doi:10.3390/ma16155376_

Round 1

Reviewer 1 Report

REVIEW

on article

Argon Plasma Treatment Effects on the Micro-Shear Bond Strength of Lithium

Disilicate with Dental Resin Cements

Yixuan Liao, Stephen Lombardo and Qingsong Yu

SUMMARY

The article submitted for review is relevant and devoted to an interesting scientific problem. The argon-plasma treatment effects on the micro-shear bond strength of lithium disilicate with dental resin cements have been studied. The authors solved an urgent problem, since dental polymer cements are a popular area in medicine, and their characteristics are the subject of study by various scientific teams.

The authors applied a deep methodology and obtained a number of important results. Experimental studies have been verified by analytical processing and, thus, the effect obtained can be applied in the practice of medicine and dentistry. At the same time, their research is new for science and brings new knowledge to scientific practice. Thus, given the scientific novelty and practical significance of the study, the reviewer proposes to support the article.

At the same time, the article has several shortcomings that need to be corrected. They are listed below.

COMMENTS

1.      The abstract is not quite correct. There is no formulation of the scientific problem. I recommend the authors to formulate the problem more specifically: what is the scientific deficit that is being filled by the research? Why it is so important?

2.      The methodology, on the contrary, is described in too much detail in the abstract. A detailed description of the methodology and materials used is very important in the second section of the article “Materials and Methods”. In Abstract, this can be reduced a bit.

3.      Finally, the scientific result should be formulated more clearly from a quantitative and qualitative point of view.

4.      I didn't see the "keywords" section. Perhaps the authors forgot to present this section for review. This section should be added.

5.      In addition, in the "Introduction" section, there is no clear transition to the purpose, tasks and problems of the study. I recommend the authors to finalize the Introduction by clearly formulated scientific gap and the main aim of the article.

6.      The author provides a review of 26 works, but this review is superficial and does not reflect the current state of the issue. The author should finalize the literature review and present it in the amount of 30–35 references, as well as supplement it with clear statements of the purpose, tasks and problems of the study.

7.      Materials and methods that are selected in section 2 need to be substantiated.

8.      Figures 3-7 look interesting, but they are poorly explained. They should be supplemented description.

9.      The SEM analysis presented in Figure 8 and the morphological images in Figure 9 also need further explanation.

10.   The graph in Figure 11 is not entirely clear, and the format chosen is rather strange.

11.   The discussion presented by the authors seems to be quite complete. I recommend the authors to add a smoother transition between discussion and conclusions, that is, the "Discussion" section should summarize the resulting part, including the findings and their analysis, as well as comparison with the results of other authors.

12.   Conclusions are presented too briefly. I would like to see a clearer view of the scientific result, scientific novelty, new knowledge gained or developed existing ideas, as well as the practical significance of the study and the prospects for its development. It is recommended to structure the conclusions and number them.

13.   I would like to wish to supplement the list of references with 5-10 references for the last five years.

14.   In general, the reviewer's conclusion is that the study is deep enough and deserves support, but some adjustments to the style of presentation and moderate corrections of English are required.

Moderate editing of English language required

Author Response

  1. The abstract is not quite correct. There is no formulation of the scientific problem. I recommend the authors to formulate the problem more specifically: what is the scientific deficit that is being filled by the research? Why it is so important?

Thanks for the suggestion. One sentence of formulating the problem was added in lines 14-15 on page 1 in the Abstract section.

  1. The methodology, on the contrary, is described in too much detail in the abstract. A detailed description of the methodology and materials used is very important in the second section of the article “Materials and Methods”. In Abstract, this can be reduced a bit.

In the revised Abstract, we have reduced the methodology by deleting some detailed description, which are given in the second section of “Materials and Methods”.

  1. Finally, the scientific result should be formulated more clearly from a quantitative and qualitative point of view.

Thanks for the suggestion. We have revised the description accordingly in the revised abstract.

  1. I didn't see the "keywords" section. Perhaps the authors forgot to present this section for review. This section should be added.

The “Keywords” section was added after the Abstract section in lines 36 and 37 on page 1.

  1. In addition, in the "Introduction" section, there is no clear transition to the purpose, tasks and problems of the study. I recommend the authors to finalize the Introduction by clearly formulated scientific gap and the main aim of the article.

Thanks for the suggestion to improve our manuscript. We have revised the Introduction section by clearly formulating the scientific gap and the main aim of article as shown in lines 70-72, 75-76, and 91-96 on page 1.

  1. The author provides a review of 26 works, but this review is superficial and does not reflect the current state of the issue. The author should finalize the literature review and present it in the amount of 30–35 references, as well as supplement it with clear statements of the purpose, tasks and problems of the study.

Thanks for the suggestions. In this revision, 32 references have been included in the Introduction section to better reflect the current state of the issue.

  1. Materials and methods that are selected in section 2 need to be substantiated.

Thanks for the suggestion. In this revision, we have revised the section by providing more details about the test specimen preparation in lines 136-139 on page 3.  

  1. Figures 3-7 look interesting, but they are poorly explained. They should be supplemented description.

Thanks for the suggestion. In this revision, more description has been added for

Figure 3: Page 6, lines 205-207

Figure 4: Page 7, lines 221-225

Figure 5: Page 8, lines 239-243

Figure 6: Page 8, lines 252-254

Figure 7: Page 9, lines 270-272 and lines 283-285

  1. The SEM analysis presented in Figure 8 and the morphological images in Figure 9 also need further explanation.

Thanks for the suggestion. In this revision, further explanation for Figure 8 and Figure 9 has been added in lines 295-308 on page 10.

  1. The graph in Figure 11 is not entirely clear, and the format chosen is rather strange.

In this revision, we have modified the figure caption for Figure 11 to better describe the data shown in the figure on Page 13.

  1. The discussion presented by the authors seems to be quite complete. I recommend the authors to add a smoother transition between discussion and conclusions, that is, the "Discussion" section should summarize the resulting part, including the findings and their analysis, as well as comparison with the results of other authors.

Thanks for the comment and suggestions. In this revision, a transition paragraph was added at the end of the discussion in lines 517-522 for a smoother transition between discussion and conclusions. In the discussion section, summary of the resulting part has been added as shown in paragraphs from lines 424-430 on page 18, lines 475-474 on page 19, lines 502-516 on pages 19 and 20.

  1. Conclusions are presented too briefly. I would like to see a clearer view of the scientific result, scientific novelty, new knowledge gained or developed existing ideas, as well as the practical significance of the study and the prospects for its development. It is recommended to structure the conclusions and number them.

Thanks for the suggestion. In this revision, we have modified the section by providing some specific scientific results in lines 532, 535-536 and the prospects for its development in lines 545-546.

  1. I would like to wish to supplement the list of references with 5-10 references for the last five years.

Thanks for the suggestion, 10 references (Ref 27, 28, 31, 32, 37, 45, 46, 47, 56, 57) from the last five years have been added in this revision.

  1. In general, the reviewer's conclusion is that the study is deep enough and deserves support, but some adjustments to the style of presentation and moderate corrections of English are required.

Thanks for the comments. In this revision, we have revised the Conclusion section by adjusting the style of presentation and some corrections of English.

Comments on the Quality of English Language: Moderate editing of English language required

Thanks for the comments. In this revision, we have done some editing to improve the quality of English writing.

Reviewer 2 Report

The manuscript “Argon Plasma Treatment Effects on the Micro-Shear Bond Strength of Lithium Disilicate with Dental Resin Cements” presents an interesting and complete study about the influence of Argon plasma treatment on lithium disilicate surfaces with effect on the micro-shear bond strength (mSBS), contact angle and failure modes of LD/resin cement. The practical applicability of the study is evident and could promote the large scale utilization of lithium disilicate by surmounting its limitations. The manuscript is well written and explicit, the research plan is logical and I recommend its publication in Materials.

Only one observation, in Figure 7, the **** are not visible, we can assume their presence, but I think they need to be added in the Figure.

Author Response

Only one observation, in Figure 7, the **** are not visible, we can assume their presence, but I think they need to be added in the Figure.

Thanks for the comments and suggestions. In this revision, we have made the **** to be clearly visible in Figure 7.

Reviewer 3 Report

This manuscript is well written, with a formulated key problem of the corresponding research object. Nevertheless, I have a few questions and comments on several aspects.

A small mistake in Fig. 6 and its description related to the representation of ** was found.

A similar problem in Fig. 7 was identified as well.

Why was the decrease of hydrophilicity of the non-thermal plasma-treated lithium disilicate surface only observed for one day? In this case, the hydrophobicity of the surface increases. It seems that after two days, it would increase even more, because after plasma treatment, its roughness increases, which means that the surface area also increases. The increase in the hydrophilicity of the lithium silicate surface after plasma treatment is related to the partial decomposition of this salt. What kind of functional groups dominate the LD surface before and after plasma treatment?

An aqueous solution of hydrogen fluoride dissolves silica and corresponding silicates, thus the statement about insoluble fluorosilicates is incorrect.

Author Response

A small mistake in Fig. 6 and its description related to the representation of ** was found.

The **** was fixed in Figure 6, with enlarged size of “****”

A similar problem in Fig. 7 was identified as well.

In this revision, we have made the **** to be clearly visible in in Figure 7.

Why was the decrease of hydrophilicity of the non-thermal plasma-treated lithium disilicate surface only observed for one day? In this case, the hydrophobicity of the surface increases. It seems that after two days, it would increase even more, because after plasma treatment, its roughness increases, which means that the surface area also increases. The increase in the hydrophilicity of the lithium silicate surface after plasma treatment is related to the partial decomposition of this salt. What kind of functional groups dominate the LD surface before and after plasma treatment?

Thanks for the comments. In this revision, we have added more discussion on the phenomena in lines 475-486 on page 19. In brief, plasma treatment increases the wettability of the LD surface by introducing more hydroxyl (OH) groups, making it more hydrophilic, through the reaction of reactive oxygen species (ROS) generated in the argon plasma with the ambient air. However, if the primer is not applied immediately after the plasma treatment, the polar groups introduced by the plasma treatment will start to react with the surrounding air. This can lead to the well know hydrophobic recovery of most of the plasma treat surfaces.

An aqueous solution of hydrogen fluoride dissolves silica and corresponding silicates, thus the statement about insoluble fluorosilicates is incorrect.

Thanks for the comments. In this revision, the inappropriate description of the HF etching of LD has been deleted. We have added references 45-47 related to HF acid etching of LD as shown in lines 452-454 on page 18.

Reviewer 4 Report

According to this manuscript, I would like to express my thanks to the authors for their efforts; it needs a major revision before evaluating the possibility of publication. I would like to pay attention to the following comments:

  • In the abstract: The objectives of the study need to be improved by addressing the used test in the study.
  • In the introduction section: The main question of the research is not addressed in the introduction section, and the introduction section also needs to be improved.
  • The introduction needs to be improved and rephrasing, such as "Introduced into the dental market in the 1990s, lithium disilicate (LD, Li2Si2O5) ceramics can be utilized both for tooth" the sentences not cleared.
  • The objectives need to clearly to be mentioned by the end of introduction section.
  • The null hypothesis should be added at the end of the introduction section and in the discussion.
  • In the methodology section, authors should add the method of sample size calculation.
  • In the discussion section, authors should explain why they choose to test the bond strength by micro-shear bond strength rather than other testing.
  • Discussion should be improved. Referring to other studies and investigations in the same field or similar to your work should be mentioned.
  • Please state some limitations of the in vitro study performed, if applicable.
  • Conclusions need to be summarized as it is too long.
  • The refences needs some updates using recent researches.

Author Response

According to this manuscript, I would like to express my thanks to the authors for their efforts; it needs a major revision before evaluating the possibility of publication. I would like to pay attention to the following comments:

  • In the abstract: The objectives of the study need to be improved by addressing the used test in the study.

Thanks for the suggestion. In this revision, we have modified the objectives of the study by addressing the used micro-shear bond strength test in lines 17-18 on page 1.

  • In the introduction section: The main question of the research is not addressed in the introduction section, and the introduction section also needs to be improved.

Thanks for the comment and suggestion. In this revision, we have modified the introduction section as detailed in lines 71-73, 76-72, and 92-99 on page 2.

  • The introduction needs to be improved and rephrased, such as "Introduced into the dental market in the 1990s, lithium disilicate (LD, Li2Si2O5) ceramics can be utilized both for tooth" the sentences not cleared.

Thanks for the suggestion. In this revision, we have modified the introduction section. The sentence of “Introduced into the dental market in the 1990s, lithium disilicate (LD, Li2Si2O5) ceramics can be utilized both for tooth” has also been edited.

  • The objectives need to clearly to be mentioned by the end of introduction section.

In this revision, we have added more sentence to clearly state the objectives shown in lines 92-99 on page 2.

  • The null hypothesis should be added at the end of the introduction section and in the discussion.

Thanks for the suggestion. The null hypothesis has been added at the end of the introduction section in lines 96-98 and in discussion section in lines 392-393 on page 17.

  • In the methodology section, authors should add the method of sample size calculation.

Thanks for the suggestion. The sample size calculation has been added in this revision in lines 137-138 on page 3.

  • In the discussion section, authors should explain why they choose to test the bond strength by micro-shear bond strength rather than other testing.

Thanks for the suggestion. In this revision, one paragraph was added to explain why the micro-shear bond strength was chosen in this study as detail in lines 507-520 on pages 19-20.

  • Discussion should be improved. Referring to other studies and investigations in the same field or similar to your work should be mentioned.

Thanks for the comment and suggestions. In this revision, several paragraphs have been added in the discussion section as shown in paragraphs from lines 424-430 on page 18, lines 475-474 on page 19, lines 502-516 on pages 19 and 20. A transition paragraph was added at the end of the discussion in lines 517-522 for a smoother transition between discussion and conclusions. 10 more references (Ref 27, 28, 31, 32, 37, 45, 46, 47, 56, 57) related to this work have been referred and added in this revision.

  • Please state some limitations of the in vitro study performed, if applicable.

Thanks for the suggestions. IN this revision, we have added the limitations of HF acid etching, thermocycling cycles number, test method shown in lines 454-457 on page 18, lines 427-433 on page 18, and lines 507-520 om pages 19-20.

  • Conclusions need to be summarized as it is too long.

Thanks for the suggestion. In this revision, we have revised the Conclusion to make the section a little shorter.

  • The refences needs some updates using recent research.

Thanks for the suggestion, 10 references (Ref 27, 28, 31, 32, 37, 45, 46, 47, 56, 57) from the last five years have been added in this revision.

Round 2

Reviewer 1 Report

All my comments were considered and corrections were done. I recommend the article for publishing.

Minor editing of English language required.